# Solitary Fibrous Tumor/Hemangiopericytoma Metastasizes Extracranially, Associated with Altered Expression of WNT5A and MMP9

**DOI:** 10.3390/cancers13051142

**Published:** 2021-03-07

**Authors:** Jong-Hwan Hong, Myung-Giun Noh, Md Rashedunnabi Akanda, Yeong Jin Kim, Se Hoon Kim, Tae-Young Jung, Shin Jung, Jae-Hyuk Lee, Joon Haeng Rhee, Kyung-Keun Kim, Sung Sun Kim, Kyung-Hwa Lee, Kyung-Sub Moon

**Affiliations:** 1Departments of Neurosurgery, Chonnam National University Research Institute of Medical Science, Chonnam National University Hwasun Hospital and Medical School, Hwasun 58128, Korea; ir16093@cnuh.com (J.-H.H.); ir12014@cnuh.com (Y.J.K.); jung-ty@chonnam.ac.kr (T.-Y.J.); sjung@chonnam.ac.kr (S.J.); 2Departments of Pathology, Chonnam National University Research Institute of Medical Science, Chonnam National University Hwasun Hospital and Medical School, Hwasun 58128, Korea; gabriel0421@gist.ac.kr (M.-G.N.); akandamr.dph@sau.ac.bd (M.R.A.); leejh@chonnam.ac.kr (J.-H.L.); succeedsoon@cnuh.com (S.S.K.); 3Department of Biomedical Science and Engineering, Gwangju Institute of Science and Technology (GIST), Gwangju 61005, Korea; 4Department of Pharmacology and Toxicology, Sylhet Agricultural University, Sylhet 3100, Bangladesh; 5Department of Pathology, Yonsei University College of Medicine, Seoul 03722, Korea; PAXCO@yush.ac; 6Medical Research Center (MRC) for Immunotherapy of Cancer, Chonnam National University Medical School, Hwasun 58128, Korea; jhrhee@jnu.ac.kr; 7Department of Pharmacology, Chonnam National University Medical School, Hwasun 58128, Korea; kimkk@jnu.ac.kr

**Keywords:** hemangiopericytoma, gene expression profiling, immunohistochemistry, neoplasm metastasis, solitary fibrous tumors

## Abstract

**Simple Summary:**

Meningeal/intracranial solitary fibrous tumor/hemangiopericytoma (*ic*SFT/HPC) have a poor clinical outcome with metastatic behavior compared to soft tissue/extracranial SFT/HPCs (*ex*SFT/HPC), but the underlying genetic factors are unclear. This study showed that WNT signaling, including *WNT5A*, was elevated in *ex*SFT/HPC and *MMP9* expression was higher in *ic*SFT/HPC at both the mRNA and protein levels. Expression of CLDN5, a marker of endothelial tight junctions, was decreased in *ic*SFT/HPC. The metastatic behavior of *ic*SFT/HPC may be due to dysregulated angiogenesis and increased permeability of the vasculature caused by an altered WNT signaling pathway. Along with the increased expression of MMP9 in individual tumor cells, the combination of these effects will increase the probability of distant metastasis. Although *ex*SFT/HPC and *ic*SFT/HPC share a key molecular event, i.e., *NAB2*-*STAT6* fusion, SFT/HPC may exhibit different biological properties and clinical courses depending on tumor location.

**Abstract:**

Solitary fibrous tumor/hemangiopericytoma (SFT/HPC) is a mesenchymal tumor originating from various soft tissues and meninges, which carries the *NAB2*-*STAT6* fusion gene. Meningeal/intracranial SFT/HPCs (*ic*SFT/HPC) have a poor clinical outcome with metastatic behavior compared to soft tissue/extracranial SFT/HPCs (*ex*SFT/HPC), but the underlying genetic factors are unclear. Differentially expressed genes (DEGs) were analyzed by NanoString nCounter assay using RNA extracted from formalin-fixed paraffin-embedded (FFPE) tissue samples. Additionally, immunohistochemistry (IHC) was performed on 32 cases of *ex*SFT/HPC, 18 cases of *ic*SFT/HPC, and additional recurrent or metastatic cases to verify the findings. Pathway analysis revealed that the WNT signaling pathway was enriched in *ex*SFT/HPC. Analysis of DEGs showed that expression of *WNT5A* was lower and that of *MMP9* was higher in *ic*SFT/HPC than in *ex*SFT/HPC (*p* = 0.008 and *p* = 0.035, respectively). IHC showed that WNT5A and CD34 expression was high in *ex*SFT/HPC (*p* < 0.001, both), while that of MMP9 was high in *ic*SFT/HPC (*p* = 0.001). Expression of CLDN5 in tumoral vessels was locally decreased in *ic*SFT/HPC (*p* < 0.001). The results suggested that decreased WNT5A expression, together with increased MMP9 expression, in *ic*SFT/HPC, may affect vascular tightness and prompt tumor cells to metastasize extracranially.

## 1. Introduction

Solitary fibrous tumor/hemangiopericytoma (SFT/HPC) is a rare mesenchymal tumor that occurs in the pleura and various extrapleural soft tissues [1]. Histologically, SFT/HPC is composed of haphazardly arranged spindle cells in a patternless architecture and is associated with variably collagenous stroma and often a staghorn-like vascular pattern [1]. Other features may include myxoid change [2], mature adipose tissue [3], floret-like giant cells (formerly referred to as giant cell angiofibroma) [4], and even dedifferentiation [5,6,7]. SFT/HPC was once considered a pericyte-originated tumor and is believed to exhibit more or less vascular-related characteristics, although it is currently subclassified as a fibroblastic tumor [1,8]. Genetic and molecular analyses of dysregulated angiogenesis have been conducted for SFT/HPC [9,10]. Vascular endothelial growth factor (VEGF) and its receptor VEGFR were upregulated [10,11] in association with activation of the AKT pathway [9,10]. These results suggested that activation of angiogenic signaling pathways is involved in tumorigenesis of SFT/HPC.

Previously, meningeal SFT and HPC were considered distinct [12,13,14]. Following the discovery of the *NAB2*-*STAT6* fusion gene in 2013 [15,16], they became a united entity based on the classification of the World Health Organization (WHO) of tumors of soft tissues [17], as was meningeal SFT/HPC in the 2016 WHO classification of the central nervous system (CNS) [18].

Although SFT and HPC were genetically integrated earlier in the extracranial soft tissues, meningeal HPCs have long been considered distinct from SFT in the intracranial location. SFT, whether it occurs in the meninges or in the extracranial soft tissues, recur very rarely and is not aggressive, but meningeal HPC frequently recurs and metastasizes extracranially [8,19]. Also, CD34 immunoreactivity is dissimilar between SFT and HPC. Almost all extra- and intra-cranial SFTs show CD34 immunopositivity (90–95%), but meningeal HPCs exhibit variable CD34 positivity rates (33–100%), typically with a patchy rather than a diffuse strong distribution pattern [8]. The WHO classification of CNS tumors uses a three-tier system that considers the SFT phenotype [19], while a two-tier system based on mitotic activity is used, irrespective of phenotype, to classify soft-tissue tumors [1].

Although integration of the tumor entities of SFT/HPC in the CNS has been accepted, previous studies to explain the causes of the difference in pathological phenotypes or tumor progression have been unsuccessful. Herein, we performed an oncogenetic analysis of SFT/HPC samples to assess differences in gene expression levels according to the location and confirmed the difference at the protein levels.

## 2. Materials and Methods

### 2.1. Sample Collection

From 2000 to 2017, cases diagnosed as SFT or HPC were retrieved from the archives of the Department of Pathology of Chonnam National University Hospital and Chonnam National University Hwasun Hospital. The initially retrieved cases were screened by STAT6 immunohistochemistry (IHC), considered a surrogate of the *NAB2*-*STAT6* gene fusion test [1], and selected by tissue availability. Fifty tissue samples of primary tumors were grouped as 18 intracranial/meningeal SFT/HPCs (*ic*SFT/HPC) and 32 extracranial/soft tissue SFT/HPCs (*ex*SFT/HPC) according to the tumor locations. The 18 *ic*SFT/HPCs included one spinal SFT/HPC. A further 13 tissue samples corresponding to extracranial metastasis and local recurrence were included. The six tissue samples corresponding to extracranial metastasis of *ic*SFT/HPC consisted of one unmatched in-house sample, three matched in-house samples, and two unmatched extra-institutional samples provided by Severance Hospital. The seven in-house tissue samples corresponding to local recurrence consisted of three matched *ic*SFT/HPC samples and four matched *ex*SFT/HPC samples, three of which were obtained from one patient.

Hematoxylin and Eosin-stained tissue slides were reviewed and re-graded by pathologists (S.S.K., J.-H.L., and K.-H.L.) according to the WHO classification of tumors of the CNS 2016 [19] and of soft tissue and bone 2020 [1]. Although one of the 18 cases of *ic*SFT/HPC showed an SFT phenotype, the grade was not indicated separately. Representative formalin-fixed paraffin-embedded (FFPE) tissue blocks were selected for further investigation. Clinical data were collected from the medical records. This study was approved by the Institutional Review Board of the Chonnam National University Hwasun Hospital (CNUHH-2017-029).

### 2.2. Gene Expression Assay Using the NanoString nCounter Analysis System

The level of gene expression was measured using the NanoString nCounter analysis system (NanoString Technologies, Seattle, WA, USA), as described previously [20]. Seven cases of *ic*SFT/HPC and six cases of *ex*SFT/HPC were subjected to the gene expression assay. For better RNA quality, the most recently acquired cases were selected from each group based on the test time point. Total RNA was extracted from 10-μm-thick FFPE sections using the High Pure FFPE RNA Isolation Kit (Roche Diagnostic, Mannheim, Germany) according to the manufacturer’s instructions. Total RNA was hybridized with preset probes and evaluated using the NanoString nCounter digital analyzer (PhileKorea Technologies, Seoul, Korea). The nCounter pan-cancer pathways panel included 770 genes from 13 canonical pathways (NanoString Technologies). Data analysis was performed using nSolver analysis software (NanoString Technologies) and the mRNA profiling data were normalized using housekeeping genes. Normalized data were log2-transformed for further analysis.

### 2.3. Pathway Dysregulation and Differentially Expressed Gene Analysis

Pathway dysregulation was scored using nSolver software (NanoString Technologies). Differential expression analysis was performed using the same regression model as the gene-level differential expression analysis. These regressions were used to calculate a *p*-value for the association of each pathway of *ic*SFT/HPC vs. *ex*SFT/HPC. Global significance statistics were also calculated for each pathway by measuring the cumulative evidence for the differential expression of genes in a pathway.

Gene set enrichment analysis (GSEA) was performed using the Molecular Signatures Database (MSigDB) v.7.0 using v. 4.0.1 of GSEA software (Broad Institute, Cambridge, MA, USA) [21,22]. The collections of MSigDB gene sets included Kyoto Encyclopedia of Genes and Genomes (KEGG) gene sets. The number of permutations was set to 1000 and the permutation type was gene set. Results were considered significant with a *p*-value < 0.05 and an FDR *q*-value < 0.1. DEG analysis was performed in nSolver Analysis Software to obtain fold differences and *p*-values. R-project v. 4.0.2 software (https://cran.r-project.org/mirrors.html; accessed date, 30 June 2020) for Mac OS was used for visualizations of DEGs.

### 2.4. Immunohistochemistry

Hematoxylin and Eosin (H&E)-stained sections were examined, and representative tissue blocks were selected for further staining. Tissue sections (3-μm-thick) were subjected to IHC using a Bond-Max Autostainer (Leica Microsystems, Buffalo Grove, IL, USA), as described previously [23]. The following antibodies were used: STAT6 (1:500 dilution; catalogue no. ab32520; Abcam, Cambridge, UK), BCL2 (1:50 dilution; M0887; DAKO, Glostrup, Denmark), CD99 (1:1000 dilution; 18-0235; Zymed, San Francisco, CA, USA), CD34 (1:400 dilution; M7165; DAKO), WNT5A (1:400 dilution; AF645; R&D, Minneapolis, MN, USA), MMP9 (1:300 dilution; #13667; Cell signaling Technology, Danvers, MA, USA), and Claudin-5 (CLDN5, 1:400 dilution; ab131259; Abcam). STAT6 staining was assessed as positive or negative to select SFT/HPC cases. The IHC scoring criteria for BCL2, CD99, and CD34 were: zero, if not expressed in any tumor cells; one, if expressed in 1–30% of the tumor area; two, if expressed in 31–60%; and three, if expressed in 61% or more. IHC results of WNT5A were scored according to staining intensity, from zero through three as none, low, moderate, and strong expression, respectively. Altered scoring criteria were applied to MMP9 IHC because only a small proportion of tumor cells was positive for MMP9. Positive staining in inflammatory cells was excluded. A score of three was assigned if there were three or more 400× high power fields with 10 or more positive tumor cells; as one, if there were fewer than five positive cells in any high-power field; as two, if a sample fell between the one-point and three-point criteria; and as zero, if not expressed in any tumor cells. CLDN5 is a marker of vascular tight junctions that shows strong expression in endothelium. Expression of CLDN5 was scored as follows: three, if expression was decreased in less than 10% of tumor vessels; two, if expression was decreased in 11–20% of tumor vessels; and one, if expression decreased in more than 21% of tumor vessels. IHC slides were assessed by experienced pathologists (S.S.K., J.-H.L., and K.-H.L.), who were blinded to the clinical details. Immunohistochemical staining was re-evaluated if there was disagreement between observers. Three pathologists reviewed the cases together and reached an agreement for inconclusive samples.

### 2.5. Immunofluorescence Staining

Immunofluorescence staining was performed as described previously [24]. Briefly, FFPE tissues of SFT/HPC cases were cut into sections 3 mm thick and deparaffinized using xylene. Heat-induced antigen retrieval was performed in citrate buffer (pH = 6.0) for 15 min, and 3% hydrogen peroxide was applied to inactivate endogenous peroxidase activity. The sections were next incubated for 16 h at 4 °C with primary antibodies targeting CD34 (anti-mouse antibody, 1:50, M7165, DAKO) and CLDN5 (anti-rabbit antibody, 1:200, ab131259, Abcam). Subsequently, sections were co-incubated with goat anti-mouse IgG (1:200, A-11001, Life Technologies, Carlsbad, CA, USA) and goat anti-rabbit IgG (1:200, A1011, Life Technologies) secondary antibodies for 1 h at room temperature. Nuclei were counterstained with 300 nM 4′,6-diamidino-2-phenylindole (DAPI) for 20 min, followed by washing with PBS. The chamber slides were mounted with antifade mounting medium and imaged using the EVOS fluorescence imaging system (Thermo Fisher Scientific, Waltham, MA, USA).

### 2.6. Statistical Analysis

The Chi-squared and Fisher’s exact tests were used to assess differences in IHC expression between *ic*SPF/HPC and *ex*SPF/HPC. To assess the association between WNT5A and MMP9, we generated a scatter plot and used the Pearson association method. The Mann–Whitney test was used to compare differences between two groups. A Kendall rank correlation analysis was performed to calculate the correlation coefficient between each variable. Statistical analysis was performed using R-project v. 4.0.2 software (https://cran.r-project.org/mirrors.html; accessed date 30 June 2020) for Mac OS. *p*-values < 0.05 were considered indicative of significance. 

## 3. Results

### 3.1. Characteristics of Selected SFT/HPC Cases

The mean ages of the patients with *ex*SFT/HPC and *ic*SFT/HPC were similar (53 and 49 years, respectively). Among the patients with extracranial tumors, the proportion of women was higher than among those with intracranial tumors (59% and 50.0%, respectively). Median tumor size was larger in *ex*SFT/HPC than in *ic*SFT/HPC (8.5 cm vs. 4.7 cm). The proportion of high histological grade was higher in *ic*SFT/HPC (9/18, 50%) than in *ex*SFT/HPC (9/32, 28%). Local recurrence was more frequently observed in icSFT/HPC (6/18, 33%) than in exSFT/HPC (3/32, 9%) and the events of distant metastasis as well (3/18, 17% vs. 2/32, 6%). The clinicopathological profiles of *ic*SFT/HPC and *ex*SFT/HPC are listed in Appendix A.

### 3.2. DEGs between icSFT/HPC and exSFT/HPC by Pan-Cancer Pathway Panel Assay

Seven cases of *ic*SFT/HPC and six of *ex*SFT/HPC were analyzed by NanoString gene expression assay. To assess the significance of 13 canonical cancer pathways in *ic*SFT/HPC compared to *ex*SFT/HPC, two statistical approaches—pathway dysregulation score and global significance statistic—were used.

Although no pathways were significantly different between *ic*SFT/HPC and *ex*SFT/HPC at *p* < 0.05, the greatest difference was in the WNT signaling pathway followed by the transcriptional mis-regulation pathway (TXmisReg) (Figure 1A). The biological significance of DEGs was explored by gene set enrichment analysis (GSEA). The WNT signaling pathway (enrichment score (ES) = 0.40, normalized enrichment score (NES) = 1.59, nominal *p* (NOM *p*)-value = 0.011 and FDR q-value = 0.344) and focal adhesion (ES = 0.39, NES = 1.49, NOM *p*-value = 0.06 and FDR q-value = 0.406) KEGG gene sets were upregulated in *ex*SFT/HPC compared to *ic*SFT/HPC (Figure 1B and Appendix A).

Of the 770 genes in the nCounter pan-cancer pathway panel, *WNT5A* was enriched in *ex*SFT/HPC with a 6.59-fold change (*p* = 0.008) and *MMP9* was enhanced in *ic*SFT/HPC with an 8659.1-fold change (*p* = 0.016) (Figure 1C). The heatmap shows the differential expression profile between *ic*SFT/HPC and *ex*SFT/HPC. Three of the ten WNT-related genes that met the significance level were overexpressed in *ic*SFT/HPC and seven were overexpressed in *ex*SFT/HPC (Figure 1D). Expression of *MMP9* was inversely correlated with that of *WNT5A WNT5A*—the higher the *MMP9* expression, the lower the *WNT5A* expression—by linear correlation analysis, but was not significant so (*p* = 0.155, *r*^2^ = 0.175; Figure 1E). The expression of *WNT5A* was significantly higher in *ex*SFT/HPC (*p* = 0.008), whereas that of *MMP9* was significantly higher in *ic*SFT/HPC (*p* = 0.035; Figure 1F). In the individual-patient precision analysis, the distance between the expression level of *WNT5A* and that of *MMP9* in *ic*SFT/HPC was greater than that in *ex*SFT/HPC (Appendix A). In the correlogram, the expression of *WNT5A* (Kendal’s τ = −0.63, *p* = 0.01) and *MMP9* (Kendal’s τ = 0.52, *p* = 0.03) showed a strong correlation with the location of SFT/HPC (Appendix A).

### 3.3. Differential WNT5A and MMP9 Protein Levels in icSFT/HPC and exSFT/HPC

To examine DEGs at the protein level, we performed IHC staining on FFPE tissues of SFT/HPC (Figure 2).

Expression of CD99 and BCL2, which were used as auxiliary diagnostic markers of SFT/HPCs prior to the use of STAT6, was not significantly different between the groups (*p* = 0.777 and 0.277, respectively). IHC of CD34 revealed diffusely strong expression in *ex*SFT/HPC but weak and patchy positivity in *ic*SFT/HPC. Expression of WNT5A was significantly higher in *ex*SFT/HPC than in *ic*SFT/HPC and expression of MMP9 was significantly enhanced in *ic*SFT/HPC (*p* < 0.001 and *p* = 0.001, respectively; Figure 3A). IHC of MMP9 revealed individual tumor cells with distinct cytoplasmic positivity. In cases with high MMP9 IHC scores, the volume of cytoplasmic positivity in tumor cells tended to become larger and the contours were more irregular. By linear correlation analysis, expression of MMP9 was inversely correlated with that of WNT5A (*p* = 0.008, *r*^2^ = 0.137; Figure 3B), similar to those at the gene level. In the correlogram, WNT5A (Kendal’s τ = −0.57, *p* < 0.001) and CD34 (Kendal’s τ = −0.72, *p* < 0.001) showed significant correlations with *ex*SFT/HPC, whereas MMP9 (Kendal’s τ = 0.50, *p* < 0.001) showed a significant correlation with *ic*SFT/HPC (Figure 3C). This finding was consistent with the DEG analysis.

### 3.4. Different Expression of CLDN5 and CD34 in icSFT/HPC and exSFT/HPC

To investigate the role of altered WNT5A expression in maintaining vascular tightness, IHC of CLDN5, a representative marker for vascular tight junctions, was performed. By IHC of CLDN5, *ex*SFT/HPC revealed well-preserved vascular tight junctions that overlapped with CD34 expression on a background of diffuse immunoreactivity in tumor cells (Figure 4A–C). Double immunofluorescence also highlighted intratumoral vessels of *ex*SFT/HPC that colocalized with CLDN5 and CD34 (Figure 4D–F). In comparison, *ic*SFT/HPCs displayed occasional foci with tumor vasculature that were negative for CLDN5 but strongly positive for CD34 (Figure 4G–I). By double immunofluorescence, some intratumoral vessels of *ic*SFT/HPCs were positive solely for CD34 and negative for CLDN5, but other vessels were positive for both CD34 and CLDN5 (Figure 4J–L). A semi-quantitative assessment showed that CLDN5 expression was significantly lower in *ic*SFT/HPCs than in *ex*SFT/HPC (*p* < 0.001, Figure 3A).

### 3.5. Reduced and Enhanced Expression of WNT5A and MMP9 in Extracranial Metastatic Lesions of icSFT/HPC

To investigate the biological role of WNT5A and MMP9 in *ic*SFT/HPC, we performed IHC on six cases of *ic*SFT/HPC with extracranial metastasis, of which four were in-house cases and two were extra-institutional cases (Appendix A). IHC of WNT5A in extracranial metastatic lesions revealed none-to-weak positivity but MMP9 IHC showed occasional tumor cells with strong cytoplasmic immunoreactivity (Figure 5A). In comparison, two cases of locally recurrent *ex*SFT/HPC maintained high expression of WNT5A and low expression of MMP9 (Appendix A). By contrast, three cases of locally recurrent *ic*SFT/HPC displayed low expression of WNT5A and high expression of MMP9, similar to metastatic lesions at extracranial locations (Appendix A).

A representative case from our archives had a disastrous clinical course caused by extracranial metastasis of *ic*SFT/HPC. Initially, the 36-year-old woman was diagnosed with grade-2 *ic*SFT/HPC and her condition was maintained without recurrence by radiotherapy (Figure 5B). The locally recurrent tumor progressed to grade-3 SFT/HPC and there was distant lung metastasis five-and-a-half years later. Despite repeated surgery and chemotherapy, the patient died from multiple lung metastases nine years after the initial diagnosis.

## 4. Discussion

We found that WNT signaling differed between *ic*SFT/HPC and *ex*SFT/HPC. Differences in WNT signaling according to SFT/HPC location influenced the angiogenesis-related pathways, which may have caused differences in the biological properties of the tumors. Among the WNT signaling genes, *WNT5A* expression was higher in *ex*SFT/HPC than in *ic*SFT/HPC.

The WNT signaling pathways produce intracellular downstream cascades in cell proliferation, differentiation, polarity, adhesion, and motility [25,26]. WNT ligands and receptors are categorized as (1) canonical WNT signaling, in which the transcriptional activity of β-catenin is a key player, and (2) non-canonical WNT signaling, which does not involve β-catenin [25]. Non-canonical WNT signaling, which regulates cell migration and polarity in major morphogenetic events including gastrulation and neural tube closure, is implicated in the regulation of angiogenesis and vascular remodeling [27,28,29]. WNT5A is a non-canonical WNT ligand that plays an important role in developmental morphogenesis —including planar cell polarity, convergent extension movements, and epithelial-mesenchymal interaction—by binding to Frizzled- and Ror-family receptors [25,30]. WNT5A is an angiogenic factor that contributes to endothelial cell proliferation and migration and vascular network formation [31,32]. Korn et al. showed that the reduced vascularization in a mouse model with deficient WNT factors could be rescued by application of WNT5A in the context of preferential expression of non-canonical WNT ligands in endothelial cells [29]. Carvalho et al. showed that WNT5A regulates angiogenic sprouting by stabilizing vinculin at cell junctions and directing cells to move collectively [33]. WNT5A was investigated in this study because we hypothesized that WNT5A is related to vascular stability in association with angiogenic property represented by CD34 immunoreactivity. We presume, however, that the non-canonical WNT11 and canonical WNT2 in the heatmap of DEGs (Figure 1D) were also related to vascular stability. WNT5A exerts both tumor-suppressing and oncogenic effects depending on the type and location of cancer [25]. WNT5A overexpression was associated with an invasive phenotype of human glioblastoma, and particularly with stem-like features in the mesenchymal subtype [34].

Tight junction integrity in endothelial cells is maintained by transmembrane proteins of the CLDN family, junction adhesion molecules and occludin, and cytosolic proteins such as the ZO family [35,36,37]. Interlinking of two extracellular loops of CLDN seals of the tight junctions between endothelial cells [38]. The expression of the CLDN family in SFT/HPC has rarely been investigated. We showed that the expression of WNT5A was low in *ic*SFT/HPC and the level of CLDN5 protein, which constitutes a tight junction, was reduced in tumor vessels in *ic*SFT/HPC. As the oncogenesis of SFT/HPC is associated with angiogenesis, maintaining the mechanical stability of tumor vessels is more difficult in *ic*SFT/HPCs than *ex*SFT/HPCs due to the decreased expression of CLDN5 and WNT5A.

MMP9 is involved in a number of biological processes, including extracellular matrix (ECM) degradation, changes of cell-to-cell or cell-to-ECM relationships, and release of ECM-sequestered molecules or cell-surface proteins [39]. In various human cancers, MMP9 is an important target and biomarker for tumor invasion, metastasis, and angiogenesis [40]. MMP9 also affects tight junctions, leading to vascular permeability in the brain [41], possibly via degradation of occludin and CLDN5 [42]. Yang et al. revealed that MMP9 degraded the tight junction proteins and induced irreversible disruption of the brain vasculature. In acute liver failure, MMP9 also degrades occludin and CLDN5 to increase vascular permeability [43]. In this study, *ic*SFT/HPCs showed higher MMP9 expression than *ex*SFT/HPCs. In *ic*SFT/HPCs, the low expression of WNT5A and high expression of MMP9 leads to low expression of CLND5. Therefore, alteration of vascular wall stability and function through WNT signaling and MMP9 in *ic*SFT/HPCs increases vascular permeability, infiltration of tissue around tumor cells, and distant metastasis. The combined effects of differential gene expression and functional alteration may explain the unusual episodes of extracranial metastasis and poor prognosis of *ic*SFT/HPCs (Figure 6).

The mean period from the initial diagnosis to extracranial metastasis in the six cases of *ic*SFT/HPC with extracranial metastasis in this study was 93 months (range, 3.8–18 years). Although only a small proportion of tumor cells showed increased MMP9 expression and tumor vessels showing decreased CLDN5 expression and low vascular tightness were distributed occasionally, extracranial metastasis likely occurs at a low frequency after a latent period of several years.

As SFT/HPC has a low prevalence, the number of tissue samples available is limited, generally in the form of FFPE. Further research involving a sufficient number of samples is needed. In addition, we performed only 770 oncogene-related gene tests in the Nanostring RNA sequencing. Whole RNA-sequencing enables more accurate DEG and gene set enrichment analyses. This study involved only genetic testing and histopathologic studies of patient tissues. The understanding of tumorigenesis would be enhanced by in vitro experiments for vascular permeation, stability, and leakage, using primary cell cultures from patient tumors.

## 5. Conclusions

Although *ic*SFT/HPC is morphologically similar to *ex*SFT/HPC and shares the *NAB2*-*STAT6* fusion gene, *ic*SFT/HPC and *ex*SFT/HPC showed differences in MMP9 expression and WNT signaling. This affects vascular stability, by impacting CLDN5 in tight junctions, causing differences in the biological behavior of *ic*SFT/HPC and resulting in more frequent metastasis and a poorer prognosis.

## Figures and Tables

**Figure 1 cancers-13-01142-f001:**
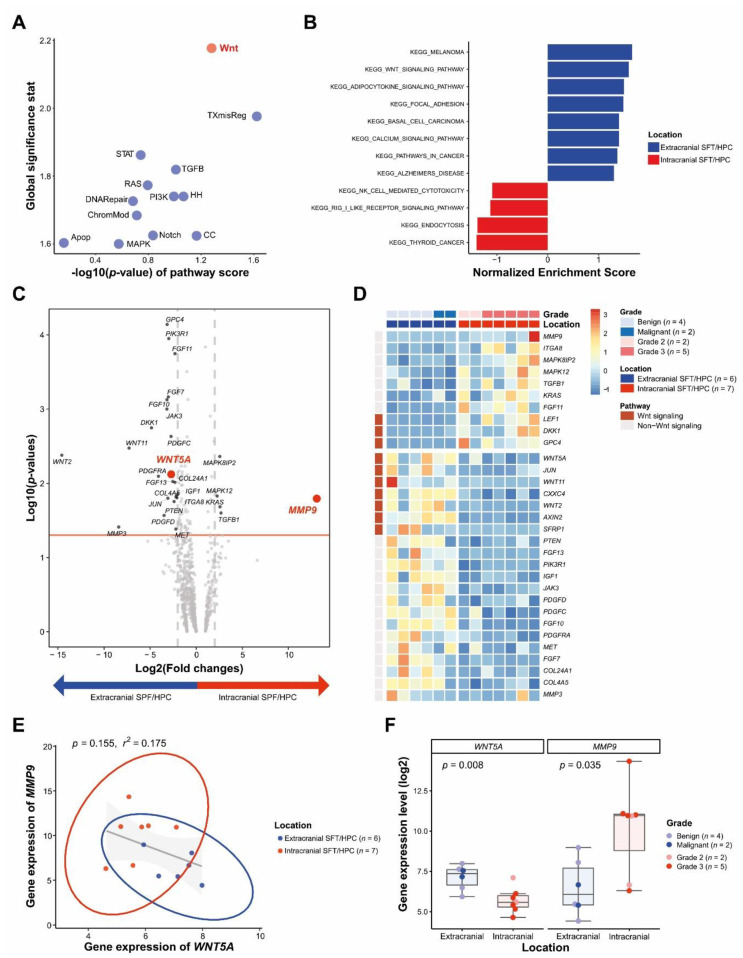
Differentially expressed genes in *ic*SFT/HPC and *ex*SFT/HPC. (**A**) Based on global significance statistics and pathway dysregulation *p*-values, the greatest difference between the two groups was observed in the WNT signaling pathway. (**B**) Gene set enrichment analysis (GSEA) showed that WNT signaling and focal adhesion pathways of KEGG gene sets were enriched in *ex*SFT/HPC compared to *ic*SFT/HPC. (**C**) Volcano plot of differentially expressed genes according to location, a fold change in expression > 2, and *p* < 0.05. (**D**) Heatmap of representative genes showing differential expression profiles in *ic*SFT/HPC and *ex*SFT/HPC. (**E**) Linear correlation analysis of *WNT5A* with *MMP9* expression. Circles represent groupings based on the location of SFT/HPC. (**F**) Box plots showing that the expression level of *WNT5A* was high in *ex*SFT/HPC and that of *MMP9* was high in *ic*SFT/HPC.

**Figure 2 cancers-13-01142-f002:**
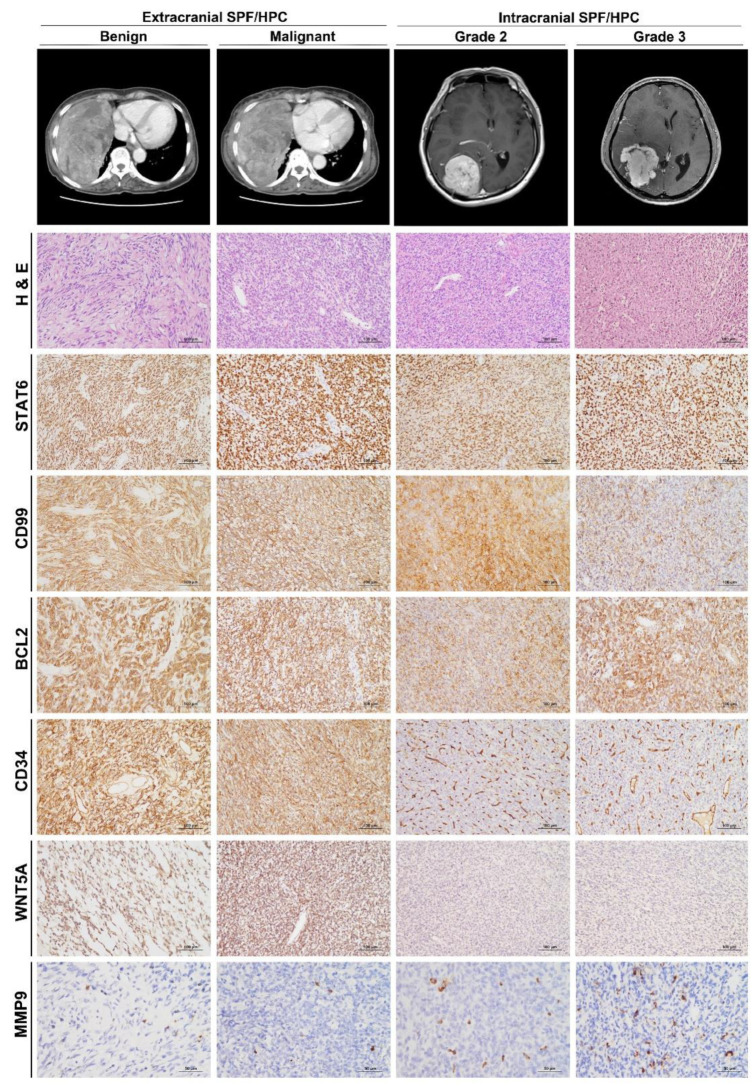
Representative images and histofigures of *ic*SFT/HPC and *ex*SFT/HPC. From top to bottom, chest computed tomography (CT), brain magnetic resonance imaging, Hematoxylin and Eosin (H&E) staining, and immunohistochemical staining for STAT6, CD99, BCL2, CD34, WNT5A, and MMP9 (original magnification: 200× for H&E, STAT6, CD99, BCL2, CD34, and WNT5A; and 400× for MMP9). From left to right, benign and malignant *ex*SFT/HPCs were followed by grade 2 and 3 *ic*SFT/HPCs. After a review of H&E slides, SFT/HPC cases were confirmatively selected by nuclear immunoreactivity of STAT6. Compared to the similar expression of CD99 and BCL2 across grades and locations, the CD34 immunostaining intensity was weaker in *ic*SFT/HPC than in *ex*SFT/HPC. Expression of WNT5A was also lower in *ic*SFT/HPC than *ex*SFT/HPC, but expression of MMP9 in occasional tumor cells was enhanced in *ex*SFT/HPC compared to *ic*SFT/HPC.

**Figure 3 cancers-13-01142-f003:**
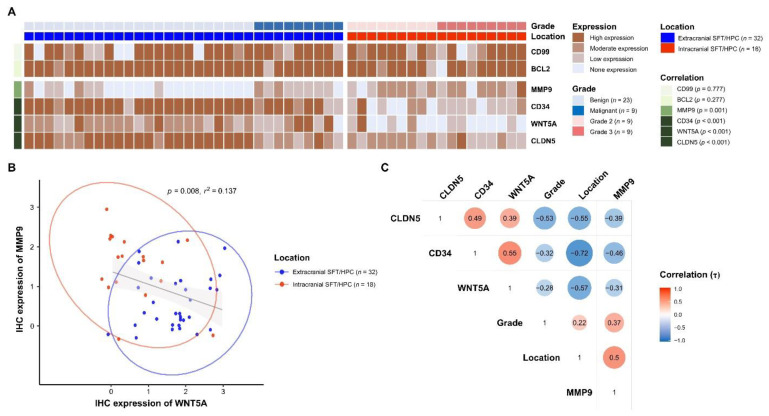
Analysis of IHC staining of SFT/HPC. (**A**) Heatmap of IHC expression of BCL2, CD99, MMP9, CD34, WNT5A, and CLDN5, displayed with grades and locations of SFT/HPCs. *p*-values are for the correlation between location and IHC expression. (**B**) Linear correlation analysis of IHC expression of MMP9 and WNT5A. Circles represent groupings according to SFT/HPC location. (**C**) Correlogram of IHC expression of CLDN5, CD34, WNT5A, and MMP9 according to location and grade of SFT/HPC. Red circles indicate a positive correlation between the paired variables and blue circles a negative correlation. The intracranial location was coded as a positive value, and the extracranial location as a negative value.

**Figure 4 cancers-13-01142-f004:**
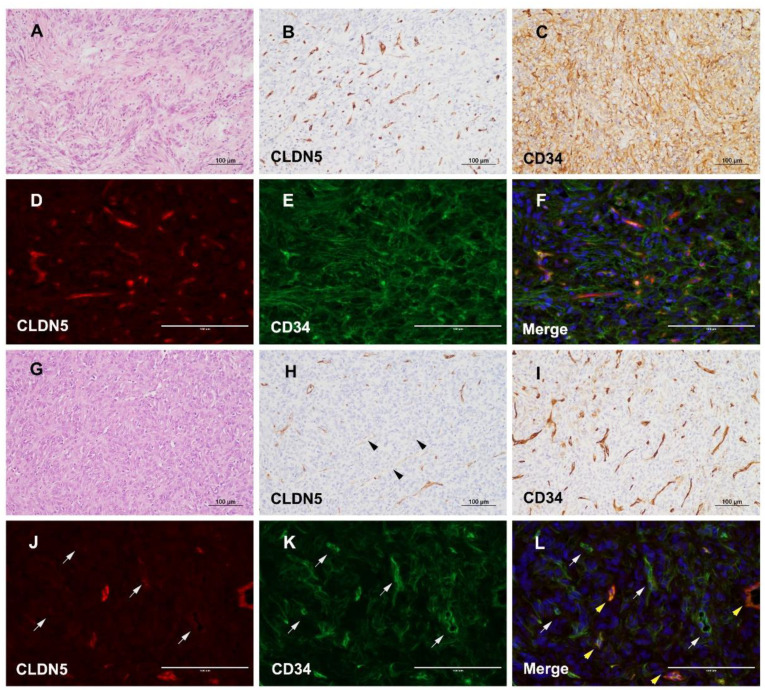
CLDN5 and CD34 expression in *ic*SFT/HPC and *ex*SFT/HPC. (**A**–**F**) Representative *ex*SFT/HPC case. H&E staining showed typical microscopic features of *ex*SFT/HPC (**A**). IHC staining showed moderate to strong expression of CLDN5, indicating well-preserved tight junctions of the tumor vasculature (**B**), and diffuse positivity for CD34 in *ex*SFT/HPC (**C**). Double immunofluorescence showed strong expression of CLDN5 (**D**), and diffuse expression of CD34 (**E**). Merged image showing tumor vessels with well-preserved coexpression of CLDN5 and CD34 in *ex*SFT/HPC (**F**) (**D**–**F**, scale bar, 100 μm). (**G**–**L**) Representative *ic*SFT/HPC case. H&E staining showed typical microscopic features of *ic*SFT/HPC (**G**). IHC showed some vessels with significantly decreased expression of CLDN5 (black arrowheads) (**H**), compared to densely labeled CD34 in tumor vessels of *ic*SFT/HPC (**I**). Double immunofluorescence revealed some tumor vessels where the expression of CLDN5 was markedly decreased (**J**) but the expression of CD34 was maintained (gray arrows) (**K**). Merged image showing good contrast between vessels stained for only CD34 (gray arrows) and other vessels with coexpression of CLDN5 and CD34 (yellow arrowheads) (**L**) (**J**–**L**, scale bar, 100 μm).

**Figure 5 cancers-13-01142-f005:**
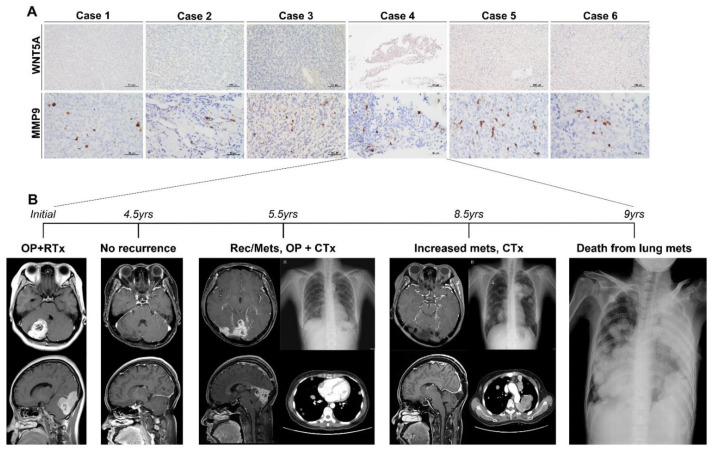
Cases of *ic*SFT/HPC exhibiting extracranial metastasis. (**A**) In all six cases, WNT5A expression was markedly decreased but that of MMP9 was significantly increased by IHC. (**B**) The clinical progression of case four was as follows: after surgical resection and radiotherapy, the patient showed no recurrence for four-and-a-half years. On follow-up imaging at five-and-a-half years postoperatively, the patient showed distant lung metastasis and local recurrence at the intracranial location. Surgical resection of the intracranial lesion and systemic chemotherapy were performed. At eight -and-a-half years after the initial operation, the metastatic lesions became widespread despite continued chemotherapy, and the patient died six months later.

**Figure 6 cancers-13-01142-f006:**
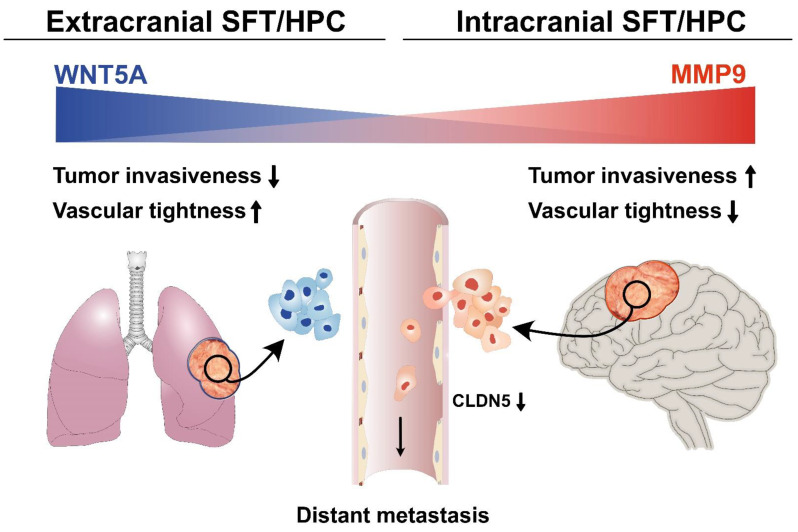
Hypothetical mechanism of extracranial metastasis of *ic*SFT/HPC. Compared to *ex*SFT/HPC, *ic*SFT/HPC shows a decreased expression of WNT5A, which maintains vascular tightness in the central nervous system and increased expression of MMP9, which promotes tumor-cell intravasation. Decreased expression of CLDN5, a marker of vascular tight junctions, in *ic*SFT/HPC is supportive of the hypothesis.

## Data Availability

The datasets generated during and/or analyzed during the current study are available from the corresponding author on reasonable request.

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
