# Peer review of "Solitary Fibrous Tumor/Hemangiopericytoma Metastasizes Extracranially, Associated with Altered Expression of WNT5A and MMP9"

_cancers, 2021, doi:10.3390/cancers13051142_

Round 1

Reviewer 1 Report

I read the study carried out by the authors on SFT/HPC tumor with great interest.
The tumor, particularly rare, shows different aggressiveness that appear to depend on the site of onset. To date, it is unclear whether there are any molecular pathways that could explain the increased risk of metastasis in inSFT/HPC.
The authors observed  molecular and immunohistochemical differences that could explain this behaviour.
The authors could insert:
- survival curves for groups.
- survival curves for WNT5A and MMP9 in inSFT and exSFT

Author Response

[ Reviewer 1]

I read the study carried out by the authors on SFT/HPC tumor with great interest.

The tumor, particularly rare, shows different aggressiveness that appear to depend on the site of onset. To date, it is unclear whether there are any molecular pathways that could explain the increased risk of metastasis in inSFT/HPC.

The authors observed molecular and immunohistochemical differences that could explain this behaviour.

The authors could insert:

- survival curves for groups.

- survival curves for WNT5A and MMP9 in inSFT and exSFT

Response) While preparing the manuscript, we once considered to provide Kaplan Meier survival curves in extracranial vs. intracranial groups. We, however, noticed that it was difficult to obtain good-quality survival data, because a majority of the exSFT/HPC patients have stopped follow-up. exSFT/HPC is generally regarded as a benign entity, and the biological properties of the tumor are actually good, follow-up observation is stopped within 1-2 years after surgery. As a result, the cases of exSFT/HPC were censored after a short follow-up (clustered in the red circle of figure A below), whereas the icSFT/HPC cases were usually followed up for a long time. So an adequate comparison of survival rates was not made (P=0.527). There was a lot of difference compared to when the authors constructed the simulation data under the assumption that exSFT/HPC patients were observed for a long period of time (P=0.003).

In addition, when the difference in survival rate according to the expression level of WNT5A was obtained in the total patient group combined with exSFT/HPC and icSFT/HPC, the trend of low survival rate in the low expression group was confirmed (figure B below). But this was also difficult to trust, since it was not acquired from high-quality survival data.

When examining the difference in survival rate according to the level of WNT5A expression by limiting to the icSFT/HPC, the group with low expression showed a significantly lower survival rate, but did not show a statistical significance due to the limitation of the number of cases (figure C below).

Taken together, the data related to the survival rate were not included into the manuscript, because they were not supposed to support the original purpose of our study, which was intended to reveal the difference in the biological properties of icSFT/HPC compared to exSFT/HPC.

Reviewer 2 Report

The following manuscript detects alteration of MMP9 and Wnt5a in solitary fibrous tumours.

The case load is limited - however with this rare disease the team showed amazing results.

Minor Revision:

I want to see the number of samples (n) in all pictures

I fully support publication after implementation of the minor change.

Author Response

The following manuscript detects alteration of MMP9 and Wnt5a in solitary fibrous tumours.

The case load is limited - however with this rare disease the team showed amazing results.

Minor Revision:

I want to see the number of samples (n) in all pictures

I fully support publication after implementation of the minor change.

Response) We have inserted the number of samples in the figures 1, 3, and supplementary 2 that are considered to require sample numbers. All the figures have been replaced with new figures.